# Effects of Perceived Food Store Environment on Malnutrition and Frailty among the Food-Insecure Elderly in a Metropolitan City

**DOI:** 10.3390/nu13072392

**Published:** 2021-07-13

**Authors:** Yu-Mi Kim, Narae Yang, Kirang Kim

**Affiliations:** 1Department of Preventive Medicine, College of Medicine, Hanyang University, Seoul 04763, Korea; kimyumi@hanyang.ac.kr; 2School of Public Health, Hanyang University, Seoul 04763, Korea; 3Department of Food Science and Nutrition, Dankook University, Cheonan 31116, Korea; skfo2581@daum.net

**Keywords:** frailty, food insecurity, supermarkets, malnutrition, aged

## Abstract

This study aimed to identify food environment factors in the local community that could affect the levels of nutritional status and frailty in 372 older adults (at least 65 years old) experiencing food insecurity and enrolled in the integrated Community Health Promotion Program (CHPP) in two districts of Seoul. The local food environment was assessed using perceived food store accessibility questionnaires. In order to quantify nutrient intake, the 24-h recall method was applied. Malnutrition was measured using the Mini Nutritional Assessment tool, while frailty was assessed using the Frailty Measurement Questionnaire developed for the CHPP. Malnourished or frail elderly adults commonly had a lower intake of cereals and potatoes, meats, and vegetables than those who were not, and their resulting intake levels of energy, protein, iron, and vitamin groups were also significantly lower (all *p*-values < 0.05). Among the local community food store environment factors, the sufficiency of food stores (odds ratio (OR) = 1.988, 95% confidence interval (CI] = 1.211–3.262), freshness of foods (OR = 1.767, 95% CI = 1.075–2.886), and variety in foods (OR = 1.961, 95% CI = 1.197–3.212) were significant factors affecting the risk of malnutrition. For frailty, the freshness of foods (OR = 1.997, 95% CI = 1.053–3.788), variety in foods (OR = 2.440, 95% CI = 1.277–4.661), and small purchase of foods (OR = 2.645, 95% CI = 1.362–5.139) were significant environmental factors. In conclusion, we found that the perceived food store environment in the local community can influence the occurrence of malnutrition and frailty in vulnerable, urban older adults.

## 1. Introduction

The elderly population (i.e., aged 65 or older) in the Republic of Korea accounted for 14.9% of the total population in 2019, showing a rapid increase from 7% in 2000. The pace of elderly population growth is the fastest among OECD countries [1]. Healthy aging requires the maintenance of physical, mental, and social capabilities and functions, thus enabling well-being in older age [2]. 

A clinically recognizable state of increased vulnerability resulting from ageing can be defined as frailty, which has been operationally defined as a syndrome characterized by weight loss, decreased grip strength, severe fatigue, slow walking speed, and a low level of activity [3,4]. In a 2018 survey conducted in a metropolitan city in the Republic of Korea, 34.6% of the elderly population was found to be frail [5]. Although limited to direct comparisons across studies due to differences in measurement methods, a systematic review reported the overall frailty prevalence as 10.7% [6]. Compared to similar East Asian countries, it is higher than 8–15% in China [7] and higher than 7.4% in Japan [8]. 

The most prominent characteristic of frailty is muscle loss [9] and, when accompanied by weight loss, improving or reversing frailty and body functions becomes very difficult. Muscle loss and weight loss have shown a close correlation with malnutrition [10], which may occur due to the decrease in quantity and quality of food consumed with age, deterioration in gustation, decreased ability to masticate by loss of dental function, gastrointestinal disorders, decreased digestive and absorption functions [11,12], and tedious food supply and eating habits following the loss of ability for social activities [13]. 

Another predisposing factor affecting health and malnutrition in the elderly is food insecurity: a state of being without reliable availability to a sufficient quantity of adequate food [14]. According to the results of the Korea National Health and Nutrition Examination Survey (KNHANES), 14.3% of the elderly experience food insecurity, with this proportion being 72.4% in the elderly belonging to the low-income population [15]. The food environment refers to the physical, economic, political, and socio-cultural environment affecting the food choices and nutritional status of individuals [16]. The local food environment has been emphasized as a factor that could influence the adequacy of food intake, food insecurity, and health status [17,18,19,20]. Previous studies have shown that the factors affecting food insecurity in the elderly are their area of residence, social isolation, low economic level, housing cost, family composition or living alone, oral health status, disability, chronic comorbidity with medication, frequency of food purchase, and difficulty in preparing food [14,15,21,22]. Recently, geographic access to grocery stores in local communities has been reported as a food environment factor affecting the food insecurity of vulnerable elderly people in the rural areas of South Korea [23,24]. The results of a domestic study on the elderly in low-income families in urban areas identified the availability of food stores around the house, physical accessibility, the reasonableness of food prices, acceptability, and convenience as factors affecting food choices [25].

Considering the effects of the food environment on healthy food choices and food insecurity, understanding the local food environment factors affecting malnutrition and frailty in the elderly is crucial for the development of community nutrition services and intervention strategies. Thus, this study aims to identify local food environment factors that may affect the levels of malnutrition and frailty in the urban elderly population from low-income families experiencing food insecurity.

## 2. Materials and Methods

### 2.1. Study Participants and Setting

The study was conducted from July to August 2018, and involved 372 subjects enrolled in the integrated Community Health Promotion Program (CHPP) in 2 districts of Seoul (Geumcheon-gu, Gangbuk-gu) and who were supposed to participate in the supplemental food assistance program, Seoul city’s newly launched program in September 2018. The CHPP provides integrated health management services to people aged 65 or older who have hypertension or diabetes, among national basic livelihood act recipients, based on nurse home visits. This service aims to administer continuous health and risk factor management, and to introduce customized local resources (e.g., community health centers, dementia support centers, mental health promotion centers, and so on). The elderly population in the 2 districts was approximately 92,000, while the CHPP was provided to approximately 13,300 older adults in the 2 districts, accounting for a coverage rate of 14.5%. Of the CHPP participants, the number of older adults who regularly visited for disease management was approximately 5600 (42.1%). Among them, the supplementary food assistance program was provided to 407 people with food insecurity, who were referred by visiting nurses of CHPP and voluntarily participated in the program. The standard for the visiting nurse’s referral was based on the guidelines of the CHPP [26], which judges whether food insecure elderly people are subject to previous national nutrition support projects. The study subjects finally amounted to 372 subjects, after eliminating those with missing variables. We obtained prior approval from the Institutional Review Board of Dankook University for all the procedures performed in this study before written consent regarding voluntary participation was obtained from the subjects (IRB no.: DKU 2018-06-001-002).

### 2.2. General Characteristics and Food and Nutrient Intake Assessment

Trained interviewers conducted face-to-face interviews based on home visits. Sex, date of birth, whether they lived alone or not, income category, the presence of three or more chronic diseases, the presence of a physical disability, cooking ability, and level of social activities were investigated.

The food intakes of subjects were measured using a 24-h recall method for the quantity of food consumed during a day. The nutrient intake was calculated using the CAN-PRO 5.0 software (Computer Aided Nutritional Analysis Program developed by the Korean Nutrition Society for comparison with RDA for Koreans) for experts, and total energy, protein, vitamin A, vitamin C, vitamin B_1_, vitamin B_2_, niacin, calcium, iron, and sodium were calculated [27]. The quantity of intake for each food group was calculated for the following categories: cereal and potatoes, legumes, meats, eggs, fish and shellfish, vegetables, fruit, milk and dairy products, and protein sources (beans, meats, eggs, fish, and shellfish). In order to evaluate the adequacy of nutrient intakes, the following Dietary Reference Intake (DRI) values for Koreans [27] were used: Estimated Energy Requirement (EER) for total energy intake, Reference Nutrients Intake (RNI) for protein, vitamin A, vitamin C, vitamin B_1_, vitamin B_2_, niacin, calcium, and iron, and Dietary Goal (DG) for sodium. 

### 2.3. Measurement of Malnutrition and Frailty 

Malnutrition was measured using the Mini Nutritional Assessment Tool, wherein a good nutritional state is represented by a score of 24 or higher, at risk of malnutrition by a score from 17 to 23.5, and malnutrition by a score of 17 or lower [28]. Frailty was assessed using the Frailty Measurement Questionnaire developed for the CHPP [26]. Scores classified the subjects into healthy (0 to 3 points), pre-frailty (4 to 12 points), and frailty (13 points or higher) groups.

### 2.4. Food Store Accessibility Environment in a Local Community

For assessment of the food environment in residential areas, a perceived food store accessibility questionnaire was developed, based on the Five A’s of food access [17,29]: Availability, affordability, accessibility, accommodation, and acceptability. The questionnaire was further revised using the results of a qualitative study which derived food environment questions applicable to the domestic, urban elderly population [25]. 

The developed questionnaire consisted of 6 items regarding the quantitative sufficiency of food stores around the house (1 item on availability), reasonableness of the cost of food sold at the food stores (1 item on affordability), delivery service availability at the food store, as well as convenience of small purchases (2 items on accommodation), variety of food sold at local food stores, and freshness (2 items on acceptability). The questions were answered on a 5-point Likert scale, consisting of Strongly agree (5 points), Agree (4 points), Neither agree nor disagree (3 points), Disagree (2 points), and Strongly disagree (1 point). A four-point or higher answer was categorized into “Positive food environment”, otherwise it was categorized into “Negative food environment” (3 points or lower). The number of positive responses for the 6 items was used as a summary measurement for the local community food store environment. 

### 2.5. Statistical Analysis 

The demographic, socioeconomic, physical, and social characteristics, food and nutrient intakes, and food store accessibility were summarized and compared in terms of malnutrition and frailty. The results of categorical variables are presented as frequencies (N) and percentages (%), while the results of continuous variables are presented as means and standard errors. A Chi-square test and independent *t*-test were performed for the univariable analysis, in order to compare the distribution of covariates and food store accessibility according to malnutrition and frailty. Multiple logistic regression analysis was conducted to determine whether food store accessibility in the local community influenced malnutrition and frailty, even after adjusting for other confounding variables. All the statistical analyses were conducted using the IBM SPSS Statistics 23.0 software (IBM SPSS INC, Armonk, NY, USA) and were tested at an alpha error level of 0.05.

## 3. Results

### 3.1. Distribution and Characteristics of Malnutrition and Frailty

Table 1 shows the demographic, socioeconomic, physical, and social characteristics affecting the malnutrition and frailty of the subjects. Of the subjects, 30.4% were men and 58.3% were 75 years old or older. Approximately 45.2% of the subjects had three or more chronic diseases, while 88% had no difficulty in moving. Approximately 25% of the subjects answered that they could not cook, and half responded that they did not engage in social activities. 

In the entire study population, 32.8% were malnourished and 67.2% were at high risk of malnutrition, with none of the subjects being in a good nutritional state. Malnutrition was higher in women than in men, and a significantly higher percentage of those aged 75 or older were malnourished. 

Frailty was observed in 14.5% of all subjects, with women showing a higher percentage than men and those aged 75 or older showing a higher percentage than those under 75 years of age. Differences in the level of frailty were not found for sex, and were only observed for age (*p* = 0.012). Subjects who had restricted mobility had a significantly higher percentage of malnutrition rather than risk of malnutrition, and a significantly higher percentage of them showed frailty rather than pre-frailty (*p* < 0.001). The subjects who could not cook showed a significantly higher percentage of malnutrition than risk of malnutrition (*p* = 0.008). There was no significant relationship between malnutrition or frailty based on living alone, household income category, number of chronic diseases, social activity, and participation in the national food support program.

### 3.2. Food and Nutrient Intake According to Malnutrition and Frailty 

Table 2 shows the distribution of the food and nutrient intake, according to malnutrition and frailty. Except for iron and sodium, the intake levels of energy and nutrients did not meet the EER and RNI for all participants. 

The food sources that showed significant differences between the levels of malnutrition were cereals and potatoes (*p* < 0.001), meats (*p* = 0.006), and vegetables (*p* < 0.001), with malnourished subjects having a lower intake of these food groups than those at risk of malnutrition. The percentage of DRI for nutrient intakes that showed significant differences among malnutrition levels were energy (*p* < 0.001), proteins (*p* < 0.001), vitamin B_1_ (*p* < 0.001), niacin (*p* < 0.001), iron (*p* < 0.001), and sodium (*p* = 0.002), with malnourished subjects consuming less of these than those at risk of malnutrition. 

Food groups that showed a difference in intake, depending on the level of frailty, were cereals and potatoes (*p* = 0.018), legumes (*p* = 0.045), meats (*p* = 0.039), fish and shellfish (*p* = 0.042), and vegetables (*p* = 0.031), with a lower intake for subjects in the frailty group than those in the pre-frailty group. In terms of the percentage of DRI for nutrient intake, the pre-frailty group showed a higher percentage of energy (*p* < 0.001), proteins (*p* < 0.001), vitamin B_1_ (*p* = 0.027), niacin (*p* < 0.001), and iron (*p* = 0.008) than the frailty group.

### 3.3. Food Store Accessibility According to Malnutrition and Frailty 

Table 3 shows the distribution of the food store accessibility perception according to malnutrition and frailty. Among food store accessibility items, a higher proportion of subjects who had a negative perception on the sufficiency of local food stores (*p* < 0.001), freshness of foods (*p* = 0.002), variety in foods (*p* = 0.002), and small purchase of foods (*p* = 0.005) were malnourished, compared to those at risk of malnutrition, than those who had a positive perception of the above. The sum of food store accessibility items was significantly lower in the malnourished subjects than that of those who were at risk of malnutrition. 

A negative perception of the sufficiency of food stores (*p* = 0.009), freshness of foods (*p* = 0.003), variety in foods (*p* = 0.001), and small purchase of foods (*p* < 0.001) showed a correlation with a higher percentage of frailty.

### 3.4. Food and Nutrient Intake across Food Store Accessibility 

The local community food store environment factors that showed relationships with the food and nutrient intakes of participants were the sufficiency of food stores, freshness of foods, and variety in foods (Table 4). Subjects who perceived that there were sufficient food stores had a higher intake of meats than those who did not, subjects who perceived that the food sold at food stores was fresh had a significantly higher intake of vegetables and fruits than those who did not, and subjects who perceived that a variety of foods was being sold had a significantly higher intake of fruits than those who did not.

As for the percentage of DRI for nutrient intake, energy, protein, niacin, and iron intake were higher in those who had a positive perception of food store sufficiency, food freshness, and food purchase in small quantities, while vitamin B_1_ intake was higher in subjects who had a positive perception of food store sufficiency and food freshness than those who did not. The percentage of DRI for Vitamin C intake was higher in those who had a positive perception of food freshness. The percentage of DRI in sodium intake was higher in the subjects who had a positive perception of reasonableness in food price, while the percentage of DRI in calcium and vitamin A intake was higher in subjects with positive perceptions of small purchases of food than those who did not.

### 3.5. Effects of Food Store Accessibility Perception on the Risk of Malnutrition and Frailty 

Some local community food store environment factors still affected the results, even after adjusting for demographic characteristics and the food and nutrient intakes affecting the levels of malnutrition and frailty (Table 5). For malnutrition, the sufficiency of food stores (odds ratio (OR) = 1.988, 95% confidence interval (CI) = 1.211–3.262), freshness of foods (OR = 1.767, 95% CI = 1.075–2.886), and variety in foods (OR = 1.961, 95% CI = 1.197–3.212) were significant environment factors affecting the risk of malnutrition. For frailty, the freshness of foods (OR = 1.997, 95% CI = 1.053–3.788), variety in foods (OR = 2.440, 95% CI = 1.277–4.661), and small purchase of foods (OR = 2.645, 95% CI = 1.362–5.139) were significant environment factors. Subjects who had a negative perception of these food store accessibility factors were at an increased risk of malnutrition or frailty, compared to those who had a positive perception of them. 

With regards to the risk of malnutrition and frailty, depending on the number of local community food store accessibility items met out of six, an increase in the number of factors decreased the risk of malnutrition (OR = 0.86, 95% CI = 0.766–0.966) and frailty (OR = 0.826 95% CI = 0.709–0.961) by approximately 14% and 17%, respectively.

## 4. Discussion 

In this study, we attempted to identify whether perceived food store accessibility, as a local community food environment factor, could affect the level of malnutrition or frailty in the vulnerable elderly population who had lower-income status and were experiencing food insecurity. The results showed that, when food store accessibility was positively perceived, the malnutrition and frailty risks decreased. Among food store accessibility factors, the ones that particularly affected the risk of malnutrition were the perceptions that there were insufficient food stores in the area, food not being fresh, or the variety of food being lacking, for which the risk of malnutrition was significantly higher. The risk of frailty significantly increased when the elderly perceived that the food sold at the food stores was not fresh, that there was not enough variety, or that making small purchases of food was not easy. 

All of the subjects in this study had experienced food insecurity and were enrolled in the CHPP as part of a national public health program. The CHPP has been criticized, as there were few nutrition services specialized to diseases and that the food provided was also not sufficient [30,31]. The food sources and nutrient intake for the subjects in this study were very low [32], reaching only half to one-third of that of the elderly in the national representative survey. Existing studies have reported that food insecurity in the elderly increases their risk of malnutrition, frailty, and chronic diseases [33], and that food insecurity itself is affected by local community food environment factors [23,24]. Areas without a variety of foods, stores, and supermarkets, as well as high food prices, have been found to affect the food insecurity of subjects [34]. Recently, local community food environment factors, such as accessibility to supermarkets, long distances between home and grocery stores, and convenience of using public transport, showed a significant correlation with food security in the Korean elderly population residing in rural areas [23]. Due to differences in the local community food environment, there was a significant difference in the dietary quality and the level of food insecurity between the elderly residing in urban areas and those residing in rural areas [35]. 

In the food-insecure elderly of this study, the risk of malnutrition or frailty was found to increase with increased negative perceptions of local community food store accessibility, which might imply that the local community food environment affected the food or nutrient intake of individuals, even for vulnerable elderly in an urban area. Previous studies have reported that the local community food environment is an important factor for food selection, quality of meals, and health outcomes [18,19,36]. In this study, the malnourished or frail elderly had a lower intake of meats, vegetables, and fruits than those who were not, and their corresponding energy, protein, iron, and vitamin intakes were also lower, as a result.

The food environment factors that have shown correlations with food selection or dietary quality include physical accessibility, reasonableness of prices, and availability of a variety of food [34,36,37,38]. In this study, the ease of small purchases was also found to be an important food environment factor influencing the intake of healthy food groups. The ease of making small purchases of food was a significant factor derived from a qualitative survey in a previous study [25]. This factor could have affected the food intake or outcome factor, as the elderly have a high rate of living alone, especially low-income elderly people. The cost burden, as well as food quantity, could have been limiting factors in food selection. The reasonableness of food prices, which has been found to be associated with the level of healthy food group intake in previous studies [29,39,40], was found to be unrelated to the nutritional status or the health outcome of the subjects in this study; among food environment factors, the percentage of positive perception on food prices within the local community was relatively higher, compared to other food environment factors. It is possible that, given the characteristics of metropolitan cities, good accessibility to many food stores could have allowed the elderly to have access to food at reasonable cost, which may have reduced the impact of food prices in relative terms. In fact, the elderly in the Republic of Korea can use public transportation (e.g., the subway) for free and, therefore, have more opportunities to purchase fresh food at reasonable prices [25]. This finding was particularly consistent with the results of overseas studies that involved the vulnerable elderly [41,42,43]. The results of these studies showed that the urban elderly with low socioeconomic levels exhibited dynamic food purchasing behavior, as they searched for grocery stores that suited their individual needs rather than going to the nearest one to purchase major food items. This study found a relationship between perceived food accessibility and malnutrition and frailty after considering several covariates, but we could not identify the variables by which perceptions of food environment were involved in the pathway and/or mechanism, leading to malnutrition and frailty due to cross-sectional design.

In this study, we examined the food environment using perceived measurements of food store accessibility. Several studies have shown that the relationship between food environment and food purchasing and intake is more significant in perceived measures than that in objective measures, especially among disadvantaged populations [23,40], suggesting that perception of access might be a better predictor of food accessibility. Other studies have reported that perceived local food environment differs by socioeconomic status [25,44,45], showing that the disadvantaged population is more likely to have negative perceptions than the less disadvantaged population. This might be explained by non-economic factors which have impacts on perceptions of food cost and availability, such as value for money, quality of produce, and convenience, as well as economic factors, such as overall budget, proportional spending, and so on [44]. These factors could account for the socioeconomic differences in food environment perception which were not attributed to the objective food environment measures. The subjects in this study were food-insecure older adults who might more negatively perceive the environment, relative to the actual food environment. Therefore, further research is required to identify the factors affecting the perception of food accessibility. These findings could help to develop intervention programs that incorporate strategies to overcome such negative perceptions. 

In this study, unmet food environment factors increased the risk of malnutrition or frailty, a finding that was still significant even after adjusting for demographic characteristics related to malnutrition and frailty, in addition to the intake of energy and food group intake related to outcome factors. This suggests that local community food environment factors could be explained by other mechanisms, in addition to food or nutrient intake. The interactions between individual and environmental factors may play important roles, in terms of the effects that the local community food environment has on eating behavior and the resulting health outcomes. For example, individual factors (e.g., self-efficacy, beliefs, and perceived disability [46,47]) or socio-environmental factors, such as the local environment (e.g., local crime rate, local deprivation index), social capital, and social networks [20,48] could interact with environmental factors to have an effect on health outcomes [43,49,50]. Therefore, in order to clearly understand the contextual effects of how the local community food environment affects malnutrition and frailty in the elderly, it would be necessary to identify the relative contributions of individual somatic and psychological factors, along with socio-environmental factors, as well as to conduct an in-depth analysis on the cross-level interactions among these factors.

The results of this study were not stratified by sex, but analyzed and presented for the entire subjects. The proportion of women in the study was large and the subjects had several chronic diseases. Because the comorbidities with consequent frailty, as well as gender, would affect the overall outcomes in the elderly [51,52], further studies with more robust design are needed to elucidate the differences between genders. 

There were several limitations in the interpretation of the results of this study. First, although the items measured in this study for the local community food environment were operationally developed based on theories and previous empirical survey tools, additional validation of the questionnaire in the survey is required. Specifically, physical accessibility was not included, as the subjects of this study lived in Seoul, which is a metropolitan city with easy physical access to local grocery stores [25]. Nevertheless, as the subjective perception of accessibility may differ from subject to subject, a survey and assessment of this factor should be conducted in the future. Second, the results of this study were cross-sectional, which made it difficult to establish causal relationships among food environment perception (e.g., food store accessibility), malnutrition, and frailty. It is necessary to clarify the causality among the factors in a future prospective study. Third, the subjects of this study were food-insecure elderly individuals in two districts of Seoul, who agreed to participate in the study, which makes it difficult to generalize the study results to other populations. The intensity of local food environment perception might vary, depending on their socioeconomic position and somatic and mental conditions. However, the results of the study, which were observed in a group with limited food purchasing power, were likely attenuated. As regional characteristics could have affected the food environment within that region, conducting a study involving a variety of regions is also necessary. 

Despite these limitations, the study identified perceived food store environment factors in a local community that could affect malnutrition and frailty in the vulnerable, urban elderly with low income, and was the first to show that these factors could affect their nutritional status and somatic condition (e.g., frailty). These results suggest the need for food support programs and nutritional intervention projects targeting the elderly, as well as the application of additional strategies to improve the local food environment, demonstrating that they could be used as foundational data for developing such strategies. Particularly, in the midst of the current pandemic, continuous monitoring is required, as the local community food environment factors are expected to have a greater impact on the nutritional status or health outcomes of the elderly. 

## 5. Conclusions

This study demonstrated that the perceived food store environment in the local community can influence the occurrence of malnutrition and frailty in the vulnerable elderly. Specifically, the sufficiency of food stores in the local community, the freshness and variety of foods, and the ease of purchasing in small quantities were identified as significant factors. In the future, these results may serve as a basis for developing food support programs and establishing nutrition policies, taking into account the food environment characteristics of the local communities wherein elderly individuals with malnutrition and frailty live.

## Figures and Tables

**Table 1 nutrients-13-02392-t001:** Distribution of malnutrition and frailty in study participants.

	All		Malnutrition					Frailty				
			Malnutrition		At Risk of Malnutrition		*p*	Frailty		Pre-Frailty		*p*
	*n* = 372		*n* = 122	(32.8%)	*n* = 250	(67.2%)		*n* = 54	(14.5%)	*n* = 317	(85.2%)	
Gender												
Male	113	(30.4)	26	(23.0)	87	(77.0)	0.008	13	(11.5)	99	(87.6)	0.290
Female	256	(69.6)	96	(37.5)	163	(63.7)		41	(16.0)	218	(85.2)	
Age												
<74	155	(41.7)	42	(27.1)	113	(72.9)	0.048	14	(9.0)	140	(90.3)	0.012
≥75	217	(58.3)	80	(36.9)	137	(63.1)		40	(18.4)	177	(81.6)	
Living alone												
Yes	285	(76.6)	90	(31.6)	195	(68.4)	0.366	43	(15.1)	242	(84.9)	0.597
No	87	(23.4)	32	(36.8)	55	(63.2)		11	(12.6)	75	(86.2)	
Household Income												
<100% of the minimum living cost	282	(75.8)	93	(33.0)	189	(67.0)	0.539	36	(12.8)	245	(86.9)	0.176
<50% of standard median household income	75	(20.2)	26	(34.7)	49	(65.3)		16	(21.3)	59	(78.7)	
Others	15	(4.0)	3	(20.0)	12	(80.0)		3	(20.0)	13	(86.7)	
Mobility												
Not good	44	(11.9)	27	(61.4)	17	(38.6)	<0.001	16	(36.4)	28	(63.6)	<0.001
Good	327	(88.1)	94	(28.7)	233	(71.3)		38	(11.6)	288	(88.1)	
Social activity												
At least once a week	144	(38.7)	48	(33.3)	96	(66.7)	0.592	16	(11.1)	127	(88.2)	0.327
Sometimes	38	(10.2)	15	(39.5)	23	(60.5)		7	(18.4)	31	(81.6)	
No	190	(51.1)	59	(31.1)	131	(68.9)		31	(16.3)	159	(83.7)	
Number of chronic diseases												
<3	204	(54.8)	67	(32.8)	137	(67.2)	0.983	27	(13.2)	177	(86.8)	0.426
≥3	168	(45.2)	55	(32.7)	113	(67.3)		27	(16.1)	140	(83.3)	
Cooking ability												
Not enable	44	(25.8)	42	(95.5)	54	(122.7)	0.008	9	(20.5)	87	(197.7)	0.095
Enable	276	(74.2)	80	(29.0)	196	(71.0)		45	(16.3)	230	(83.3)	

Data are expressed as *n* (%), *p*-values were obtained by Chi-square test.

**Table 2 nutrients-13-02392-t002:** Food sources and nutrient intake according to malnutrition and frailty.

	All	Malnutrition			Frailty		
		Malnutrition	At Risk of Malnutrition	*p*	Frail	Pre-Frail	*p*
	*n* = 372	*n* = 122 (38.2%)	*n* = 250 (67.2%)		*n* = 54 (14.5%)	*n* = 317 (85.2%)	
Food sources (g)							
Cereals and potatoes	204.7 ± 5.6	171.3 ± 6.7	221 ± 7.5	<0.001	172.4 ± 11.3	210.2 ± 6.3	0.018
Legumes	39.8 ± 4.2	38.7 ± 7.2	40.3 ± 5.1	0.856	24.9 ± 6.9	41.8 ± 4.7	0.045
Meats	21.6 ± 2.3	13.7 ± 2.8	25.5 ± 3.2	0.006	12.6 ± 4.4	23.2 ± 2.6	0.039
eggs	15.6 ± 1.7	16.4 ± 3.1	15.2 ± 2.0	0.727	10.7 ± 4.0	16.5 ± 1.9	0.238
Fish and shellfish	18.2 ± 2.0	15.2 ± 3.4	19.7 ± 2.5	0.294	11.8 ± 2.7	19.1 ± 2.3	0.042
Vegetables	168.3 ± 6.2	126.1 ± 8.6	188.9 ± 8.0	<0.001	135.5 ± 13.6	173.8 ± 6.9	0.031
Fruits	53.7 ± 5.7	42.3 ± 8.5	59.2 ± 7.4	0.136	32.6 ± 13.6	57.4 ± 6.3	0.101
Dairy	48.5 ± 5.6	64.5 ± 10.1	40.8 ± 6.7	0.051	45.1 ± 12.3	49.3 ± 6.3	0.791
Nutrient intakes as a percentage of DRI (%)							
Energy	56.6 ± 21.3	49.0 ± 18.3	60.3 ± 21.7	<0.001	46.6 ± 17.8	58.3 ± 21.4	<0.001
Protein	68.9 ± 36.0	57.9 ± 31.5	74.2 ± 36.9	<0.001	52.5 ± 25.9	71.6 ± 36.8	<0.001
Vitamin A	31.7 ± 38.2	29.0 ± 37.8	33.1 ± 38.4	0.333	23.6 ± 22.6	33.2 ± 40.2	0.091
Vitamin C	31.2 ± 32.1	27.4 ± 30.2	33.0 ± 32.9	0.115	27.1 ± 32.0	31.9 ± 32.1	0.312
Vitamin B1	72.4 ± 38.7	60.2 ± 35.4	78.4 ± 38.9	<0.001	61.6 ± 37.9	74.2 ± 38.6	0.027
Vitamin B2	47.5 ± 32.3	44.0 ± 31.9	49.2 ± 32.4	0.147	39.7 ± 29.6	48.8 ± 32.6	0.056
Niacin	39.2 ± 22.2	32.3 ± 19.3	42.6 ± 22.7	<0.001	29.5 ± 15.0	40.8 ± 22.8	<0.001
Ca	33.6 ± 23.8	30.6 ± 20.2	35.1 ± 25.3	0.092	28.1 ± 19.3	34.5 ± 24.4	0.065
Fe	103.5 ± 58.1	83.6 ± 45.3	113.3 ± 61.2	<0.001	84.3 ± 45.7	106.8 ± 59.5	0.008
Na	101.5 ± 65.5	86.9 ± 62.4	108.6 ± 65.9	0.002	92.1 ± 56.4	102.8 ± 66.7	0.266

Data are expressed as mean ± standard deviation, *p*-values were obtained by independent *t*-test for the difference in nutrient intakes, compared to Dietary Reference Intakes (DRI) between nutrition or frail status.

**Table 3 nutrients-13-02392-t003:** Univariate relationship of food store accessibility perception with malnutrition and frailty.

	All	Malnutrition				Frailty			
		Malnutrition		At Risk of Malnutrition		Frail		Pre-Frail	
	*n* = 372	*n* = 122 (38.2%)		*n* = 250 (67.2%)		*n* = 54 (14.5%)		*n* = 317 (85.2%)	
Sufficiency of food stores									
Sufficient	185(49.7)	44	(23.8)	141	(76.2)	18	(9.7)	167	(90.3)
Not sufficient	187(50.3)	78	(41.7)	109	(58.3)	36	(19.3)	150	(80.2)
*p*		<0.001				0.009			
Freshness of foods for sale									
Fresh	192(51.6)	49	(25.5)	143	(74.5)	18	(9.4)	174	(90.6)
Not fresh	180(48.4)	73	(40.6)	107	(59.4)	36	(20.0)	143	(79.4)
*p*		0.002				0.003			
Variety in foods for sale									
Variety	193(51.9)	49	(25.4)	144	(74.6)	17	(8.8)	176	(91.2)
No variety	179(48.1)	73	(40.8)	106	(59.2)	37	(20.7)	141	(78.8)
*p*		0.002				0.001			
Price of foods for sale									
Affordable	227(61.0)	76	(33.5)	151	(66.5)	35	(15.4)	191	(84.1)
Not affordable	145(39.0)	46	(31.7)	99	(68.3)	19	(13.1)	126	(86.9)
*p*		0.725				0.525			
Easy to purchase in small packages									
Easy	191(51.3)	50	(26.2)	141	(73.8)	15	(7.9)	176	(92.1)
Not easy	181(48.7)	72	(39.8)	109	(60.2)	39	(21.7)	141	(78.3)
*p*		0.005				<0.001			
Service of food stores									
Good	113(30.4)	33	(29.2)	80	(70.8)	13	(11.5)	100	(88.5)
Not good	259(69.6)	89	(34.4)	170	(65.6)	41	(15.9)	217	(84.1)
*p*		0.33				0.27			
Number of positive responses for food store accessibility items									
Mean ± SD	3.0 ± 2.0	2.5 ± 2.0		3.2 ± 1.9		2.1 ± 2.0		3.1 ± 2.0	
*p*		0.001				0.001			

Data are presented as *n* (%), *p*-values were obtained by Chi-square test and by independent *t*-test for sum of items.

**Table 4 nutrients-13-02392-t004:** Food and nutrient intake, according to food store accessibility perception.

	Sufficiency of Food Stores			Freshness of Foods for Sale			Variety in Foods for Sale			Price of Foods for Sale			Easy to Purchase in Small Packages			Service of Grocery Store		
	Not Sufficient	Sufficient	*p*	Not Fresh	Fresh	*p*	No Variety	Variety	*p*	Not Affordable	Affordable	*p*	Not Easy	Easy	*p*	Not Good	Good	*p*
Food sources (g)																		
Cereals and potatoes	197.7 ± 8.7	211.8 ± 7.2	0.209	201.4 ± 9.2	207.8 ± 6.7	0.570	202.8 ± 9.4	206.4 ± 6.5	0.750	210.9 ± 10.1	200.7 ± 6.6	0.379	202.0 ± 9.2	207.3 ± 6.7	0.639	204.6 ± 7.3	204.8 ± 8.0	0.991
Legumes	39.7 ± 5.7	39.8 ± 6.1	0.984	35.7 ± 5.5	43.6 ± 6.2	0.341	40.1 ± 6.0	39.4 ± 5.8	0.938	34.6 ± 6.1	43.0 ± 5.6	0.324	39.4 ± 5.7	40.1 ± 6.1	0.928	38.3 ± 4.8	43.1 ± 8.1	0.599
Meats	16.4 ± 2.6	26.9 ± 3.9	0.026	17.1 ± 2.8	25.9 ± 3.7	0.056	20.0 ± 3.1	23.1 ± 3.5	0.512	22.6 ± 3.4	21.0 ± 3.2	0.730	17.7 ± 2.7	25.4 ± 3.7	0.100	23.4 ± 3.1	17.5 ± 3.0	0.173
Eggs	15.7 ± 2.6	15.4 ± 2.2	0.931	15.1 ± 2.6	16.0 ± 2.3	0.808	16.5 ± 2.6	14.8 ± 2.2	0.618	14.2 ± 2.6	16.4 ± 2.3	0.532	16.9 ± 2.5	14.3 ± 2.3	0.460	15.7 ± 2.1	15.4 ± 3.1	0.932
Fish and shellfish	19.1 ± 3.1	17.4 ± 2.6	0.674	19.9 ± 3.2	16.6 ± 2.4	0.409	19.6 ± 3.2	17.0 ± 2.5	0.510	15.5 ± 2.7	20.0 ± 2.8	0.276	16.5 ± 2.8	19.8 ± 2.8	0.408	17.2 ± 2.5	20.5 ± 3.4	0.454
Vegetables	154.2 ± 8.1	177.7 ± 9.1	0.056	153.3 ± 8.1	177.7 ± 9.1	0.047	155.8 ± 8.2	175.3 ± 9.0	0.113	174.0 ± 10.5	160.7 ± 7.5	0.290	156.1 ± 8.4	175.1 ± 8.9	0.122	170.0 ± 7.6	156.5 ± 10.4	0.311
Fruits	42.7 ± 7.7	64.7 ± 8.4	0.055	39.2 ± 7.5	67.2 ± 8.5	0.014	39.8 ± 7.1	66.5 ± 8.8	0.019	44.5 ± 7.4	59.5 ± 8.1	0.171	47.1 ± 7.6	59.9 ± 8.5	0.265	51.5 ± 6.8	58.6 ± 10.6	0.570
Dairy	51.0 ± 7.7	46.0 ± 8.1	0.659	48.4 ± 7.5	48.6 ± 8.3	0.985	46.5 ± 7.5	50.5 ± 8.3	0.723	45.1 ± 7.9	50.7 ± 7.7	0.630	47.0 ± 7.3	50.0 ± 8.5	0.789	48.3 ± 7.2	49.1 ± 8.2	0.949
Nutrient intakes as a percentage of DRI (%)																		
Energy	53.3 ± 21.4	59.9 ± 20.8	0.003	53.6 ± 21.4	59.4 ± 20.8	0.009	54.7 ± 21.6	58.3 ± 20.9	0.108	55.5 ± 17.9	57.3 ± 23.2	0.404	53.5 ± 21.0	59.5 ± 21.2	0.007	56.3 ± 21.8	57.1 ± 20.1	0.729
Protein	63.5 ± 34.0	74.2 ± 37.3	0.004	64.2 ± 35.2	73.2 ± 36.3	0.016	66.0 ± 34.5	71.5 ± 37.2	0.136	64.9 ± 29.2	71.4 ± 39.6	0.070	63.4 ± 32.3	74.1 ± 38.6	0.004	68.8 ± 38.2	69.0 ± 30.6	0.948
Vitamin A	28.8 ± 34.7	34.7 ± 41.4	0.140	28.0 ± 34.3	35.3 ± 41.4	0.067	28.6 ± 34.2	34.6 ± 41.5	0.129	30.6 ± 35.0	32.5 ± 40.3	0.635	27.7 ± 34.7	35.6 ± 41.0	0.046	32.0± 39.8	31.2 ± 34.6	0.863
Vitamin C	28.3 ± 27.7	34.1 ± 35.8	0.077	26.6 ± 25.3	35.5 ± 36.9	0.006	28.4 ± 26.5	33.8 ± 36.4	0.109	29.7 ± 26.6	32.1 ± 35.1	0.463	28.3 ± 28.2	33.9 ± 35.2	0.097	31.6 ± 30.8	30.3 ± 34.9	0.733
Vitamin B1	67.7 ± 37.4	77.2 ± 39.5	0.018	67.9 ± 37.4	76.7 ± 39.4	0.028	70.3 ± 38.7	74.3 ± 38.7	0.319	74.1 ± 39.1	71.4 ± 38.4	0.507	68.6 ± 38.5	76.1 ± 38.6	0.063	72.2 ± 39.5	72.9 ± 36.8	0.869
Vitamin B2	44.7 ± 31.3	50.3 ± 33.1	0.091	44.4 ± 29.6	50.4 ± 34.5	0.071	46.2 ± 29.8	48.7 ± 34.4	0.464	45.4 ± 26.7	48.9 ± 35.4	0.278	44.4 ± 29.5	50.4 ± 34.5	0.075	47.1 ± 32.5	48.4 ± 31.8	0.716
Niacin	35.8 ± 21.1	42.7 ± 22.7	0.002	36.5 ± 21.8	41.8 ± 22.2	0.022	37.4 ± 21.6	40.9 ± 22.6	0.134	39.3 ± 23.6	39.2 ± 21.2	0.967	35.0 ± 18.6	43.2 ± 24.5	<0.001	38.4 ± 21.5	41.1 ± 23.7	0.280
Ca	32.1 ± 24.8	35.2 ± 22.7	0.215	31.4 ± 23.9	35.7 ± 23.6	0.085	32.2 ± 23.7	34.9 ± 23.9	0.271	31.9 ± 22.6	34.7 ± 24.5	0.259	31.1 ± 23.3	36.0 ± 24.1	0.047	33.7 ± 25.5	33.5 ± 19.5	0.934
Fe	97.0 ± 61.6	110.2 ± 53.6	0.028	95.6 ± 57.1	111.0 ± 58.2	0.011	98.7 ± 59.4	108.0 ± 56.7	0.121	101.8 ± 56.1	104.7 ± 59.4	0.637	96.5 ± 57.4	110.2 ± 58.1	0.023	104.0 ± 61.4	102.6 ± 50.1	0.837
Na	95.0 ± 60.0	108 ± 70.1	0.056	98.0± 65.9	104.8 ± 65.0	0.320	98.3 ± 65.1	104.5 ± 65.8	0.362	93.2 ± 54.3	106.8 ± 71.3	0.037	96.6 ± 64.3	106.1 ± 66.4	0.164	100.9 ± 66.0	102.9 ± 64.6	0.792

Data are expressed as mean ± standard deviation, *p*-values were obtained by independent *t*-test for the difference in nutrient intakes, when compared to Dietary Reference Intakes, between the perception of food store accessibility status.

**Table 5 nutrients-13-02392-t005:** Effects of food store accessibility perception on malnutrition and frailty.

	Malnutrition				Frailty			
	Model 1		Model 2		Model 1		Model 2	
	aOR	(95% CI)	aOR	(95% CI)	aOR	(95% CI)	aOR	(95% CI)
Sufficiency of food stores								
Sufficient	1.000		1.000		1.000		1.000	
Not sufficient	2.257	(1.407–3.618)	1.988	(1.211–3.262)	1.936	(1.034–3.625)	1.698	(0.894–3.225)
Freshness of foods for sale								
Fresh	1.000		1.000		1.000		1.000	
Not fresh	1.985	(1.242–3.171)	1.761	(1.075–2.886)	2.175	(1.163–4.071)	1.997	(1.053–3.788)
Variety in foods for sale								
Variety	1.000		1.000		1.000		1.000	
No variety	2.040	(1.277–3.258)	1.961	(1.197–3.212)	2.492	(1.321–4.699)	2.440	(1.277–4.661)
Price of foods for sale								
Affordable	1.000		1.000		1.000		1.000	
Not affordable	0.934	(0.584–1.495)	0.964	(0.588–1.579)	0.770	(0.411–1.441)	0.793	(0.419–1.503)
Easy to purchase in small packages								
Easy	1.000		1.000		1.000		1.000	
Not Easy	1.679	(1.059–2.661)	1.481	(0.912–2.404)	2.944	(1.533–5.654)	2.645	(1.362–5.139)
Service of food store								
Good	1.000		1.000		1.000		1.000	
Not good	1.169	(0.706–1.935)	1.187	(0.700–2.013)	1.355	(0.679–2.706)	1.322	(0.654–2.673)
Number of positive responses for food store accessibility items	0.835	(0.747–0.933)	0.860	(0.766–0.966)	0.803	(0.690–0.934)	0.826	(0.709–0.961)

aOR, Adjusted Odds Ratio; CI, Confidence Interval; Model 1 was adjusted for gender, age, mobility, and cooking environment, for malnutrition, it was adjusted for age, and adjusted for mobility for frailty. Model 2 was adjusted for total energy, fruits, vegetables, and meats, in addition to those of Model 1.

## Data Availability

The data used came from the CHPP and restrictions apply to the availability of these data. Data might be however available from the corresponding author upon reasonable request and with permission of the CHPP.

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
