# Peer review of "Effects of Perceived Food Store Environment on Malnutrition and Frailty among the Food-Insecure Elderly in a Metropolitan City"

_nutrients, 2021, doi:10.3390/nu13072392_

Round 1

Reviewer 1 Report

Thank you for the opportunity to review your paper, which describes the effects of perceived food store environment on malnutrition and frailty among community dwelling adults in metropolitan Korea. Your paper addresses an interesting topic and has a good sample size. However, I think the discussion may need to be refocused as you measured perceptions of the food environment among food insecure elderly adults not their actual food environment. Further to this, all subjects resided in the same two districts and received the same program (CHPP) which is meant to provide nutrition services, yet I don’t think many were compliant with/used the program. Therefore, perhaps differences in perceptions (and therefore malnutrition/frailty) were due to other factors vs. being a direct result of the environment? I’ve provided further feedback below.

Title: I think this could be changed to “The effects of perceived food store environment on malnutrition and frailty among food-insecure elderly adults in a metropolitan city”. I think “in community” can be gauged from this.

Abstract:

Line 17 - Delete ‘the’ at beginning of sentence.

Lines 18/18 – change to “Undernourished or frail elderly adults commonly...” so take out “the” and add “adults”.

Line 21 – “low” should be “lower”.

Overall: Can intake, perception and p-values be added to compare groups? There is also no mention of how food intake was measured.

Introduction

Lines 31-32 – this sentence needs to be revised. What is meant by “pathological levels”.

Line 34 – Revise the beginning of this sentence as frailty is not the only physiological vulnerability of aging.

Line 40 – I believe the most common criteria of frailty are weakness and slow gait speed. However, I see muscle loss is stated as the most prominent characteristic of frailty. If this is the case, a reference is needed to support this statement.

Lines 46-49 – Please revise this sentence, particularly the beginning – I’m unsure what you’re trying to convey. It also does not set up the next paragraph very well.

Lines 59-60- Change to “The general elderly population’s need for community nutrition services is ….” And the next sentence (lines 60-61) should either be incorporated into this sentence with “and” or “for example” instead of “meanwhile” – are they not the same thing or at least related? And the sentence also needs to be revised as it does not make sense (I think it’s the “to”).

Line 64 – remove the full stop that has been placed mid-sentence.

Overall: 1) It is very specific to Korea. I understand this is the context where the study was conducted but Nutrients is an international journal, hence, comparing/contrasting prevalence rates to international literature in the opening paragraph would be beneficial.  2) I think there needs to be greater justification as to why this study took place. What’s the gap? The last two paragraphs begin setting this up, but I think it could be strengthened.  

Methods

Lines 89-90 – Revise sentence. Potentially change to: “… older adults who were visiting regularly for disease management was about 5,600 (42.1%).”

Lines 90-911 – How did the nurses decide they were priority recommended?

Line 93 – Revise. I think you’re trying to say of the 407 high priority participants indemnified by CHIPP nurses, 372 completed the survey. Is there information anywhere as to why the other 35 people did not complete the survey?

Line 98 – “In” not “on the home”.

Line 129 – If starting a sentence with a numeral, it needs to be spelled out.

Lines 134-136 – revise as this sentence does not make since.

Overall: 1) Tense is inconsistent - methods should always be written in past tense. 2) careful with terminology – the word survey and interview appear to be used interchangeably. 2) How many questions did the food accessibility consist of? Also, if you developed it, this should be added as a supplementary document as it is currently unclear what was actually assessed. I had trouble interpreting your results in section 3.4 as I did not understand what participants were being asked. 3) It would be to include a sentence somewhere stating exactly how many assessments were undertaken and when, and information around how long they were part of the CHPP and supplemental food assistance programs before assessment took place.

Results

Lines 152 – 153 – Did you record the reasons why they did not participate in the food support program? Also, was this in reference to CHPP or the supplemental food assistance program? Please try to use consistent terminology to orientate your reader.

Lines 155 -156 – Use “were” not “was undernourished”.

Lines 158-159 – Revise sentence. “showing a higher than” does not make sense.

Lines 190-193 – Revise sentence. The first half is very confusing and provides little info.

Lines 195-196 – How does the local community food store environment relate to food source intake? Do you mean participant intake?

Line 209 – The word “perception” needs to be captured in this title (e.g. 3.5. Effects of food store accessibility perceptions on the risk of malnutrition and frailty) otherwise it may be a bit misleading.

Table 2 – Are all the food sources expressed in grams? Is dairy expressed in grams. I wonder if you can try to provide relative context to this (e.g. what is the recommended intake for each of these food sources and energy and nutrient per day?).

Discussion

Line 241 – how did you define “good” though?

Line – 245 – replace “high” with “higher”.

Line 249 – Instead of saying “none of them had a good nutritional status” perhaps use the definitions set up in the methods (malnourished or at risk of malnutrition).

Lines 249-245 – Much of this info belongs in the methods section (lines 249 -252). If included there, this sentence could be much shorter here. Also, is the intention of this program to fully support participants or to just provide aid?

Lines 333-337 – Please be sure to include the word “perceptions” otherwise it’s misleading. I’m unsure if you should be jumping to

Overall: 1) Undernourished and malnourished are used interchangeably. I would choose and use only one to avoid confusion. 2) I’m having trouble working out if participants’ perceptions are based on actual differences in their food environment or differences in their socioeconomic position / mentality. I see this is addressed in the limitations section (lines 328 -333) but I think it would benefit as being included in the main discussion. What would it mean if their food environments are the same, it’s just differences in their perceptions? What implications would this have? For this reason, I think it’s important to perhaps explain the term “district” (I saw participants were from two districts but what is the radius of a district?). Further, I think this is important as in the title you’ve described that all participants are food-insecure elderly adults, which makes me lean towards differences in their perceptions vs. actual food-environment (or compliance with the CHPP). Also, I see lines 334-343 discuss overall implications of your findings but perhaps you should discuss the need for further foundational work first (e.g. a. cross-analysis of perceptions and actual environment and b. a needs assessment of some kind to identify the gap in services/resources).

Reviewer 2 Report

In this cross-sectional study performed in 372 elderly subjects (>65 y) from two districts in Seoul, Yu-Mi Kim et al. observed that perceived food store environment in the local community can influence the occurrence of malnutrition and frailty in the vulnerable elderly.

The paper is interesting and quite original. However, this reviewer raises some issues that the authors have to addressed.

1- The manuscript presents several limitations. In particular the cross-sectional design, as correctly stated also by authors. This issue needs to be better addressed in the discussion, not simply in the limitation section.

2- The population studied is divided according to the number of chronic diseases (<3 vs >3). Moreover, the gender is not equally represented (female were about 70%). Actually, comorbidities with consequent frailty as well as the gender affect the overall outcome in the elderly, as well demonstrated in the REPOSI study (Eur J Intern Med. 2014 Sep; 25 (7): 617-23. doi: 10.1016/j.ejim.2014.06.027 - J Gerontology, Series A Biol Sci and Med Sci. 2017; 72 (3): 395-402. doi: 10.1039/gerona/glw188). This important point should be commented upon in the discussion.

3- The paper needs a linguistic revision by a native English speaker.

Author Response

I, on behalf of the authors, appreciate the reviewers for providing these insights.

We have incorporated these suggesting throughout the revised manuscript.

The major corrections were highlighted in yellow in the manuscript (nutrients-1270917_revised.docx).

An English correction and editing were conducted by MDPI English editing services.

#2-1- The manuscript presents several limitations. In particular the cross-sectional design, as correctly stated also by authors. This issue needs to be better addressed in the discussion, not simply in the limitation section.

: Since this study was designed as a cross-sectional, the considerations derived were further described in discussion part (line 311-315).

#2-2- The population studied is divided according to the number of chronic diseases (<3 vs >3). Moreover, the gender is not equally represented (female were about 70%). Actually, comorbidities with consequent frailty as well as the gender affect the overall outcome in the elderly, as well demonstrated in the REPOSI study (Eur J Intern Med. 2014 Sep; 25 (7): 617-23. doi: 10.1016/j.ejim.2014.06.027 - J Gerontology, Series A Biol Sci and Med Sci. 2017; 72 (3): 395-402. doi: 10.1039/gerona/glw188). This important point should be commented upon in the discussion.

: We agree with you and have incorporated your suggestion (line 349-352).

#2-3- The paper needs a linguistic revision by a native English speaker.

: An English correction and editing were conducted by MDPI English editing services.

Round 2

Reviewer 1 Report

Well done for making all the changes. 

Author Response

Thanks a lot.

Reviewer 2 Report

In my previous review I wrote: "The population studied is divided according to the number of chronic diseases (<3 vs >3). Moreover, the gender is not equally represented (female were about 70%). Actually, comorbidities with consequent frailty as well as the gender affect the overall outcome in the elderly, as well demonstrated in the REPOSI study (Eur J Intern Med. 2014 Sep; 25 (7): 617-23. doi: 10.1016/j.ejim.2014.06.027 - J Gerontology, Series A Biol Sci and Med Sci. 2017; 72 (3): 395-402. doi: 10.1039/gerona/glw188). This important point should be commented upon in the discussion.".

Actually, only the first reference was added and commented in the revised manuscript. I advice that authors add and comment on both references. 

Author Response

We added comments and the reference, as you suggested. (line 352-356)